# Surgical Management of Neuroendocrine Tumours of the Pancreas

**DOI:** 10.3390/jcm9092993

**Published:** 2020-09-16

**Authors:** Regis Souche, Christian Hobeika, Elisabeth Hain, Sebastien Gaujoux

**Affiliations:** 1Department of Digestive, Minimally Invasive & Oncologic Surgery, Montpellier University Hospital Centre, University of Montpellier, 641 avenue du Doyen Gaston Giraud, 34090 Montpellier, France; fr-souche@chu-montpellier.fr; 2Department of Hepato-Biliary and Pancreatic Surgery and Liver Transplantation, AP-HP Pitié-Salpêtrière Hospital, 47-83 Avenue de l’Hôpital, 75013 Paris, France; hobeikachristian@hotmail.com; 3Department of Digestive, Hepato-biliary and Endocrine Surgery, University of Paris, Hôpital Cochin–Pavillon Pasteur, 27 rue du Faubourg Saint Jacques, 75014 Paris, France; elisabet.hain@gmail.com; 4Department of General, Visceral, and Endocrine Surgery, Pitié-Salpêtrière Hospital, AP-HP, 75013 Paris, France; 5Medicine Faculty, Sorbonne University, 15-21 Rue de l’École de Médecine, 75006 Paris, France

**Keywords:** pancreatic neuroendocrine tumour, multidisciplinary team meeting, pancreatic sparing surgery, pancreatectomy, lymph node metastasis, lymphadenectomy

## Abstract

Neuroendocrine tumours of the pancreas (pNET) are rare, accounting for 1–2% of all pancreatic neoplasms. They develop from pancreatic islet cells and cover a wide range of heterogeneous neoplasms. While most pNETs are sporadic, some are associated with genetic syndromes. Furthermore, some pNETs are ‘functioning’ when there is clinical hypersecretion of metabolically active peptides, whereas others are ‘non-functioning’. pNET can be diagnosed at a localised stage or a more advanced stage, including regional or distant metastasis (in 50% of cases) mainly located in the liver. While surgical resection is the cornerstone of the curative treatment of those patients, pNET management requires a multidisciplinary discussion between the oncologist, radiologist, pathologist, and surgeon. However, the scarcity of pNET patients constrains centralised management in high-volume centres to provide the best patient-tailored approach. Nonetheless, no treatment should be initiated without precise diagnosis and staging. In this review, the steps from the essential comprehensive preoperative evaluation of the best surgical approach (open versus laparoscopic, standard versus sparing parenchymal pancreatectomy, lymphadenectomy) according to pNET staging are analysed. Strategies to enhance the short- and long-term benefit/risk ratio in these particular patients are discussed.

## Highlights

Surgical management of pNETs should be planned in a multidisciplinary staff meeting.The initial accurate assessment is the cornerstone in pNETs management and should include accurate localisation, grading, and staging.Surgery should be performed for asymptomatic non-functional pNETs > 2 cm or non-functional symptomatic pNETs regardless of tumour size.Surgery should be performed for all functional sporadic pNETs, except those with unresectable distant metastasis.Parenchyma-sparing surgery is recommended for insulinoma and can be considered for non-functional pNETs < 2 cm if associated with lymph node picking.

## 1. Introduction

Neuroendocrine tumours of the pancreas (pNET) develop from pancreatic islet cells. They are rare and their incidence is very low (1–2 tumour/100,000 inhabitants/year) [1], accounting for 1–2% of all pancreatic neoplasms. They can be diagnosed at a localised stage or more advanced stage, including regional or distant metastases (in 50% of the cases), which are mainly located in the liver.

The recent increased incidence of pNETs, especially regarding early staged tumours, probably lies in the development of cross-sectional imaging during the past decades [1,2]. The terminology of “pNET” covers a wide range of heterogeneous lesions. First, pNETs can be dichotomised according to their sporadic or genetic origin. While most pNETs are sporadic, some are associated with genetic syndromes such as Multiple Endocrine Neoplasia type 1 (MEN1), Von Hippel-Lindau disease [VHL), neurofibromatosis 1 (NF1), or tuberous sclerosis complex. Secondly, pNETs are usually associated with the secretion of peptides, being called “functioning” tumours when they are associated with clinical symptoms related to an hypersecretion of metabolically active peptides [3,4].

Nowadays, functioning pNETs account for less than 20% of cases, mainly including insulinoma, gastrinoma, and more rarely, glucagonoma, VIPoma, and somatostatinoma. As most (about 65–85%) pNETs are non-functioning, they are frequently diagnosed incidentally. Consequently, pancreatic incidentalomas represent 80% of all resected non-functioning pNETs (NF-pNETs) in recent surgical series [5,6,7]. Finally, pNETs have variable prognoses according to the differentiation grade and mitotic index/Ki67 that stratify tumours according to three grades (G1 to G3).

While surgical resection is the cornerstone of the curative treatment of pNET patients, pNET management requires a multidisciplinary team including an oncologist, radiologist, pathologist, and surgeon with substantial expertise. In this setting, the scarcity of pNET constrains a centralised management in high-volume centres (Figure 1).

## 2. Preoperative Evaluation of pNET Patients

An exhaustive preoperative evaluation is the prerequisite of pNET patient management, including clinical examination, biological tests, and radiological imaging (Figure 2). The clinical examination is dedicated to the investigation of symptoms related to the hypersecretion of active peptides, which should be corrected prior to surgery. Moreover, some hypersecretion syndromes related to functional pNET, such as insulinoma, gastrinoma, or VIPoma, could lead to life-threatening complications, therefore requiring immediate management. Elements of personal and familial medical history that could suggest the presence of a genetic syndrome (MEN1, VHL, NF1) must be sought. Moreover, most pNET are nowadays “non-functioning” and usually asymptomatic. Symptomatic pNETs (abdominal pain, tenderness and/or with an abdominal mass) are associated with a poorer prognosis [1,7,8,9,10].

The biological assessment of pNETs patients is useful for the diagnosis and monitoring of their response to treatment and surveillance. Preoperative Chromogranin A (CgA) dosage is recommended for follow-up, while it is not taken into consideration for surgical indication. Plasmatic CgA should be interpreted with caution since various clinical situations can independently increase levels, such as decreased renal and/or cardiac function, underlying liver and/or inflammatory bowel and/or autoimmune and/or chronic obstructive pulmonary diseases. Moreover, treatment with proton pump inhibitors (PPI) or somatostatin analogues also influences the plasmatic CgA level, with increased CgA level related to PPI persisting up to 2–3 weeks following the end of the treatment. The plasmatic dosage of neuron specific enolase (NSE), pancreastatin, pancreatic polypeptide (PP) or other peptides is not routinely recommended. However, the plasmatic dosage of pro-insulin, insulin, C-peptide, fasting levels of gastrin, vasoactive intestinal peptide, glucagon, and somatostatin are required in the evaluation of functional pNETs according to the clinical symptoms [11].

Radiological assessment plays a key role in the surgical management of patients, providing an accurate staging of pNETs. It encompasses multiple modalities, including computed tomography (CT), magnetic resonance imaging (MRI), endoscopic ultrasound (EUS) and nuclear medicine imaging. In CT, which should be performed using a triphasic injection (i.e., arterial, late arterial, and venous phases with thin reconstructed slices), pNETs appear as a well-limited intrapancreatic lesion and hyper-enhanced on the arterial phase. Pancreatic MRI provides additional information regarding the pNET stage and refines the assessment of the margin between the tumour and the pancreatic duct, which is paramount in the case of conservative pancreatic resection. Hepatic MRI with diffusion-weighted sequences helps detect liver metastasis [12,13,14].

Somatostatin receptor scintigraphy is useful in the preoperative management of pNETs [15], with 68 Ga-DOTATOC PET/CT being superior to the previously widely used Octreoscan^®^ with a sensitivity and specificity over 95% for pNETs detection [12,16]. 18 FDG-PET has potential, in addition to somatostatin receptor scintigraphy imaging, to assess pNETs aggressiveness and grade [16,17]. Other nuclear medicine modalities are currently in development, such as glucagon-like peptide-1 receptor (GLP-1R) PET/CT for the diagnostic and localisation of insulinoma (Figure 3) [18,19].

In the same way, an invasive technique is available in the case of occult insulinoma as a selective arterial calcium stimulation test (sensitivity 85%) in order to regionalize the insulinoma and plane either a distal pancreatectomy or a pancreaticoduodenectomy [20,21,22,23,24]. Nevertheless, this technique, at least in Europe, is more infrequently performed and rarely available even in expert centers.

EUS is an invasive and operator-dependent procedure that requires general anaesthesia. Nevertheless, it remains a useful tool to refine tumour margins with the main pancreatic duct. Needle-biopsy under EUS provides histologic grading according to the WHO 2017 classification, while pathological evidence is not necessarily required prior to surgical management. Moreover, intraoperative ultrasonography is routinely performed, especially in the case of small lesions difficult to locate during the surgical procedure [25]. In this setting, intraoperative US helps to guide the surgical strategy [i.e., non-conservative versus conservative pancreatectomy) [26].

## 3. Indications of Surgical Resection of pNET Patients

Surgical indication depends on the type of pNET [non-functional or functional, sporadic or genetic) and the preoperative tumour staging. Briefly, except for patients with G3 tumours and/or extrahepatic metastasis [addressed in a separate section), all pNET patients should be evaluated for surgical resection. While surgical resection is systematically indicated in patients with functional pNET and NF-pNET ≥2 cm, a surveillance strategy is recommended in patients with small NF-pNET or small non-metastatic MEN1 gastrinoma [27,28,29,30,31,32].

The decision to perform surgical resection in patients with pNET should be based on the recommendation of a multidisciplinary team with substantial expertise. For example, the nationwide French network of pNETs RENATEN [Réseau National de Référence pour la prise en charge des Tumeurs neuro-Endocrines) is dedicated to centralising and standardising the management of these patients. Figure 4 summarises the main surgical indications for pNETs.

### 3.1. Sporadic pNETs

#### 3.1.1. Non-Functional pNETs

The surgical indication of sporadic non-functional pNETs is guided by the clinical symptoms and the size of the lesion. Asymptomatic patients, with NF-pNETs with a maximal diameter < 2 cm without any associated dilatation of bile and/or main pancreatic duct, are eligible for simple surveillance [11,33,34,35], whereas for tumours ≥ 2 cm or associated with symptoms and/or ductal dilatation, pancreatic resection is the standard treatment.

#### 3.1.2. Insulinomas

Insulinomas are the most frequent functional pNETs with an incidence of four cases per millions of inhabitants per year [36]. Most insulinomas are sporadic and benign, with malignant lesions in less than 10% of cases.

Insulinoma is suggested in patients with severe and iterative episodes of hypoglycaemia occurring especially during the postprandial period, fasting or physical exercise. Also, these patients can have associated confusion, behavioural changes and/or vision disorders, so require prior exclusion of other causes of hypoglycaemia, confirmed biologically using plasmatic dosage and fasting tests. Insulinoma patients have low plasmatic glucose, inappropriately elevated plasmatic insulin and C-peptide [11,37]. The fasting test requires strict medical surveillance.

The control of insulinoma hypersecretion and its consequences is the priority in the management of these patients since hypoglycaemic episodes can be life threatening. In this setting, insulinoma patients require appropriate diet [frequent small meals) and adequate education regarding their diseases, the signs of hypoglycaemia and its treatments. Indeed, diazoxide and long-acting somatostatin analogues (SSAs) are efficient treatments for reducing the number and the severity of hypoglycaemic episodes. Nevertheless, while 30–50% of the patients are good responders, these treatments can exacerbate the symptomatology in some cases [38,39,40,41,42,43,44].

Surgical resection is systematically indicated for patients with insulinoma [11,33]. Because they are most of the time benign lesions, conservative pancreatectomy is preferentially indicated [45]. However, insulinomas are very small lesions that can be difficult to locate in the pancreas, therefore, extensive pre- and intraoperative explorations are required to optimise surgical management. As an alternative to surgery, endoscopic ultrasound-guided ablation therapies with ethanol or by radiofrequency have recently been developed, with limited available data to date [46,47,48].

#### 3.1.3. Gastrinomas

The diagnosis of gastrinomas is based on the presence of Zollinger-Ellison syndrome, which is characterised by the association of severe peptic ulceration and profuse volumogenic diarrhoea that is related to the hypersecretion of gastrin [11,37]. In this setting, the control of this hypersecretion using PPI (in some cases with very high doses) is the priority in the management of these patients.

Typically, gastrinomas are small and multiple lesions even in the context of sporadic cases. They are preferentially located in the so-called “Stabile and Passaro” triangle, which is an anatomic triangle defined by the junction of the cystic and common bile ducts superiorly, the junction of the second and third portions of the duodenum inferiorly, and the junction of the neck and body of the pancreas medially. Since sporadic gastrinomas can be malignant lesions, surgery, either local excision or pancreaticoduodenectomy with formal lymphadenectomy, is required and is associated with patient overall survival [49,50].

#### 3.1.4. VIPoma

VIPoma is a functional pNET responsible for WDHA syndrome, which includes watery secretory, diarrhoea, hypokaliemia, and hypo- or achlorhydria. This syndrome is the consequence of the hypersecretion of vasoactive intestinal peptide. Approximately 70% of patients are reported to have synchronous metastasis at the time of diagnosis. Therefore, VIPoma patients are associated with poor prognosis [51].

Symptoms, characteristics and surgical approach in relation with the most frequent functional pNET are summarized in Table 1 [37,52,53].

### 3.2. pNETs Occurring in MEN1 Patients

MEN1 is a rare disease with an estimated prevalence of 1 to 10/100,000 inhabitants. The diagnosis of MEN1 is suggested in the context of young patients with multiple pNETs, or pNETs associated with hypercalcaemia or another endocrinopathy at any age. Multifocal pNETs occur in more than 75% to 90% on a MEN1 background, with pNETs representing one of the most frequent cancer-related causes of mortality in MEN1 patients. Therefore, the prognosis of these patients is closely related to the management of their pNETs. However, despite various published guidelines [37,54,55,56], the management of MEN1 related pNETs patients is difficult to standardise. This lies in the fact that MEN1 related pNETs are usually multiple with various types of functional or non-functional tumours in the same patient. In this setting, achieving an accurate preoperative identification and characterisation of all pNETs of the MEN1 patients is difficult.

#### 3.2.1. MEN1 Related NF-pNETs

In the case of MEN1 related NF-pNETs, the management is similar to the sporadic cases, hence, simple surveillance is usually recommended for small (< 2 cm) and asymptomatic lesions [57,58,59], while surgical resection is recommended for larger or symptomatic lesions.

#### 3.2.2. MEN1 Related Insulinoma

Insulinomas occur on a background of MEN1 in less than 5% of the cases [60,61,62]. The surgical indication of MEN1 related insulinomas is systematic. Nevertheless, the distinction of the insulinoma(s) from other NF-pNET lesions in the context of MEN1 is challenging [63,64]. In this setting, non-conservative surgery is mainly preferred in the case of MEN1 related insulinoma associated with other pNETs.

#### 3.2.3. MEN1-Related Gastrinoma

For MEN-1-related gastrinomas, their management is mainly influenced by their size and metastatic status. Indeed, while PPI treatment is very efficient to downregulate gastrin hypersecretion, MEN1 related gastrinomas are usually multiple and located in the duodenum, hence, it is unlikely to achieve curative resection using conservative pancreatic surgery (such as enucleation] [50,65,66]. In this setting, a “wait and see” policy is usually recommended for non-metastatic small lesions (< 2 cm), while a non-conservative pancreatic surgery is indicated for larger lesions [67] and/or gastrinoma with nodal involvement [68].

### 3.3. G3 pNETs

Patients with G3 pNETs represent approximately 10–20% of all pNETs patients and are associated with poor survival [19]. The WHO 2017 classification distinguishes two categories of G3 pNETs according to their differentiation: well-differentiated G3 pNETs and poorly differentiated G3 lesions also called neuroendocrine carcinomas [69]. While curative surgery seems to improve the prognosis of patients with G3 well-differentiated pNETs, the benefit of surgery in patients with localised pancreatic neuroendocrine carcinomas is unclear [70,71]. Therefore, the decision for surgery is made on a case-by-case basis. In the case of patients with metastatic G3 pNETs carcinomas, there is no indication for surgical resection regarding the very poor prognosis of these patients.

### 3.4. Metastatic pNETs

Liver metastases occur in approximately 40–45% of pNETs and do not constitute an absolute contraindication to surgical resection [72]. Indeed, provided an extensive evaluation showing the absence of other metastatic sites except for the liver, the indication of surgical resection is related to the pNET grade and the resectability of the liver metastasis.

An upfront surgical approach is recommended in the presence of pNET G1-G2 with resectable liver metastases and discussed in the case of G3 pNETs [73,74,75]. In the presence of unresectable liver metastases, debulking palliative surgery can be considered, especially in the context of life-threatening and/or obstructive complications (bleeding, acute pancreatitis, jaundice, or gastric obstruction), even if this remains controversial. In this context, the indication of an upfront pancreaticoduodenectomy will preclude most interventional radiology procedures for the treatment of liver metastasis due to the high risk of liver abscesses and should be avoided.

## 4. Modalities of Pancreatic Resection of pNETs

### 4.1. Standard Surgery Versus Pancreatic Sparing Surgery

Standard pancreatic resection with lymph nodes dissection, including pancreaticoduodenectomy and distal pancreatectomy, is the standard non-conservative surgical treatment for pancreatic tumours such as pNETs. However, these are extensive surgical procedures associated with non-negligible morbidity and an impaired pancreatic exocrine (9–60% of cases) and endocrine function (7–35%) according to both the extent and the side of pancreatic resection [76,77]. Pancreas-sparing procedures are conservative surgical treatments and include central pancreatectomy and enucleation. In the context of benign and/or non-aggressive tumours, such as G1-G2 pNETs, these procedures preserve exocrine and endocrine pancreatic functions, providing acceptable oncological results [11]. Indeed, pancreatic sparing strategies are indicated for patients with small and clearly delineated pNETs, without any consensually defined cut-off of tumour size. Nevertheless, it seems that below a tumour size of 2 cm, oncological results of pancreatic sparing strategies are satisfactory [78,79,80,81,82] (Figure 5).

Among pancreatic sparing strategies, enucleation is indicated for superficial tumours that did not involve the main pancreatic duct and central pancreatectomy is indicated for tumours located in the body of the pancreas without the involvement of the gastroduodenal artery or contact with the common bile duct. Regarding the morbidity related to the presence of two pancreatic sections in central pancreatectomy (see above), the surgeon should consider the length of the preserved pancreatic tail [at least 5 cm) and the actual pancreatic function of the patient [presence and severity of diabetes) in the benefit–risk ratio of central pancreatectomy against distal pancreatectomy.

### 4.2. Surgical Approach: Minimally Invasive Versus Open Pancreatectomy

During previous decades, the minimally invasive approach has gained wide acceptance for the treatment of digestive tumours [83,84]. Profuse literature argues in favour of the laparoscopic approach, which is associated with reduced intraoperative blood loss, decreased postoperative pain, complication and hospital stay [85]. Altogether, the laparoscopic approach improves postoperative rehabilitation of patients providing at least similar if not better oncological results [86,87]. Minimally invasive pancreatic surgery is carefully and more slowly developed in relation to the substantial technical difficulty of the procedures, especially for pancreatic reconstruction via a laparoscopic approach (i.e., Whipple procedure). In this setting, there is currently no consensus regarding the indications of the laparoscopic approach for pancreatic resection [88,89].

The published evidence failed to show the superiority or even equivalence of the laparoscopic approach to the Whipple procedure [90]. Indeed, a randomised controlled trial was stopped prematurely because of an excess of mortality in the laparoscopic approach compared with the open approach [91]. Laparoscopic pancreaticoduodenectomy (LDP) presents some disadvantages compared with conventional laparotomy, including instrument motion with restricted range of movement, two-dimensional imaging, poor surgeon ergonomics, and a long learning curve, especially to perform pancreatic and biliary anastomosis. Robotic surgery is an advanced minimally invasive surgical technique that has several benefits in complex procedure as pancreaticoduodenectomy, such as enhanced three-dimensional vision, application of EndoWrist instruments that mimics the latter’s hands and consequently, a shorter learning curve compared to standard laparoscopy. Therefore, Robot-assisted pancreaticoduodenectomy (RAPD) might be more advantageous than LPD. The absence of reconstruction required in distal pancreatectomy makes robot-assisted distal pancreatectomy less attractive and less cost effective, especially for benign or borderline lesion as pNETs [92,93]. Recent reports have shown that RAPD is safer and more efficient than LPD among properly selected patients [94,95,96]. Therefore, RAPD is technically a feasible alternative to the laparoscopic procedure. Further studies may be needed to evaluate the cost-effectiveness of RAPD [96].

Otherwise, the results appear more encouraging for minimally invasive pancreatectomies that do not require reconstructions, explaining that laparoscopic enucleation and distal pancreatectomy were widely adopted. While the level of evidence remains poor, the literature suggests that both surgical approaches provide similar short- and long-term results, even for malignant lesions [59,85,97,98]. Moreover, laparoscopic distal pancreatectomy could be associated with a decreased rate of pancreatic fistula.

In this setting, the laparoscopic approach has been progressively developed and proposed for distal pancreatectomy and pancreatic sparing procedures in patients with NF-pNETs or clearly delineated and isolated insulinoma in centres with substantial expertise in both laparoscopic and pancreatic surgery [99,100,101]. Nevertheless, these favourable results regarding the laparoscopic approach require confirmation by further dedicated studies and randomised controlled trials.

### 4.3. Short- and Long-Term Results of Pancreatic Resection in pNETs Patients

According to the American Cancer Society (cancer.org), relying on information from the SEER* database between 2009 and 2015, the five-year survival rate for resected patient with non-metastatic pNETs is 93%. In locally advanced pNETs or positive regional lymph nodes, the five-year survival rate is 77%. In metastatic patients, the five-year survival rate is 27%.

In the context of pNETs patients who have mainly benign or non-aggressive tumours with excellent long-term prognosis, the quality of the surgical care lies on both the short and long-term outcomes and the risk of postoperative diabetes. However, pancreatic resections are associated with important postoperative morbidity related to the substantial risk of pancreatic fistula and its consequences [80].

In Whipple procedures and distal pancreatectomy, the rate of pancreatic fistula ranged from 15% to 30% of cases [102]. Pancreatic sparing procedures are associated with an increased risk of pancreatic fistula related to the presence of two pancreatic sections in central pancreatectomy and the proximity of the surgical section with the main pancreatic duct in enucleation [80]. In this setting, pancreatic sparing surgery can be associated with increased postoperative rates of complications, 25–70% for middle pancreatectomy and 43–45% for enucleation [78,79]. In comparison, the Whipple procedure and distal pancreatectomy were associated with up to 50% and 30% of overall complications rates, respectively [102].

The main advantage of pancreatic sparing procedures is to decrease the risk of pancreatic insufficiency, especially the risk of postoperative diabetes [0% in case of enucleation and usually below 10% in case of central pancreatectomy), while pNET patients who underwent surgical resection usually have a good prognosis [102,103]. In contrast, this risk of diabetes increased to 30% in Whipple procedure and distal pancreatectomy. Nevertheless, this advantage regarding the preservation of pancreatic functions has to be counterbalanced by the increased morbidity of these conservative procedures in the benefit/risk balance of the patient [6,104]. As an example, the conservative procedure is likely to benefit patients with good performance status and preserved pancreatic function, while in patients with increased perioperative risk and insulin-requiring diabetes, non-conservative procedures are safer.

### 4.4. Pancreatic Resection of Functional pNETs

#### 4.4.1. Gastrinoma

Gastrinomas are multiple tumours, which can be very small and located in the duodenum or less frequently, the pancreas, within the Stabile and Passaro triangle. In this setting, preoperative evaluation is sometimes not conclusive. Intraoperative exploration is paramount and very useful to refine the localisation of small duodenal and pancreatic lesions. It includes intraoperative fibroscopy with duodenal transillumination and bi-digital palpation for the detection of duodenal lesions and intraoperative ultrasonography for pancreatic tumours [105]. In this setting, the minimally invasive approach is not recommended because laparoscopic extensive intraoperative exploration is not possible.

Small submucosal tumours (1–2 mm of diameter) can be resected by duodenotomy [106,107], whereas larger and transmural duodenal lesions require elective duodenectomy. Typically, intrapancreatic lesions are eligible for enucleation but a non-conservative pancreatectomy, usually a pancreaticoduodenectomy, could be performed for multiple localisations. A large regional lymph node dissection must be systematically associated with the surgical procedure. Pathological examinations of the surgical sample must be performed by an experienced pathologist. Whether pancreaticoduodenectomy or enucleation/lymphadenectomy should be systematically performed remains an unsolved question.

#### 4.4.2. Insulinoma

The long-term results of pancreatic resection for insulinoma are excellent, with a 98–100% cure rate for sporadic non-malignant lesion [10,108,109,110]. Most insulinomas are benign lesions. Therefore, conservative and non-invasive treatments are preferred. Several studies showed a significant benefit from the laparoscopic approach in regard to short-term outcomes [99,100,101]. A two- or three-millimetre margin width between the tumour and the main pancreatic duct is required to safely perform enucleation.

Indeed, indications for non-conservative pancreatectomy for insulinoma are scarce. In the case of unresectable insulinoma by pancreatic sparing procedures, percutaneous or endoscopic ablations have been reported to be efficient [48,111].

Patients with malignant insulinoma are rare. Their management requires a radical and oncological approach using non-conservative pancreatectomy with appropriate lymphadenectomy.

### 4.5. The Value of Lymphadenectomy in pNETs Surgery

The techniques of pancreatic sparing strategies raised the question of lymphadenectomy in tumours which are admittedly not very aggressive. In this setting, lymphadenectomy should be systematically considered with the exception of insulinomas (without any suspicion of malignity). Several studies have reported the prognostic value of positive nodal status in pNETs patients [112,113,114,115], which account for 10% of small (<2 cm) NF-pNETs patients.

In the case of G1 tumours eligible for pancreatic sparing surgery with the absence of nodal invasion in preoperative imaging, a lymph node picking is recommended. In this setting, 68 Ga-DOTATOC PET/CT is useful to refine the localisation of the picking [16,116]. In the case of pNETs that require non-conservative pancreatectomy, an adequate lymphadenectomy (at least 12 harvested nodes) is recommended to provide an accurate staging of the patient [115].

## 5. Recurrence Rate and Follow-Up after Resection

Despite a 5-year survival rate for resected patient is around 90%, recurrence rate is 12–25% at five years and late recurrence were described (up to 70% at 15 years) [117,118,119,120,121]. Follow-up requires a close monitoring with physical examination, biology (essentially for F-pNETS) and cross-sectional imaging (CT/MRI), 3–6 monthly for NF-pNETs and 6–12 monthly for F-pNETs every year for three years, every 1–2 years thereafter, and until 10 years post-pancreatectomy. Although insulinoma may not require any radiological follow-up, pNETS guidelines recommend a life-long surveillance for all other pNETs [11,74].

## 6. Conclusions

Surgery is the cornerstone of the curative multimodal treatment of pNETs and should be tailored according to the tumour characteristics, such as accurate localisation, grading, and staging, as well as the health history of the patient. The management of pNETs should be discussed in a dedicated multidisciplinary staff meeting. Despite observation being generally recommended for small and asymptomatic pNETs usually demonstrating indolent behaviour, surgical resection is systematically indicated in patients with functional pNETs and NF-pNETs ≥ 2 cm. Well-selected patients with advanced/metastatic or well-differentiated G3 neoplasms can benefit from multimodal treatment including surgery. Further dedicated studies are required. However, the laparoscopic approach remains the gold standard for distal pancreatectomy and pancreatic sparing procedures in patients with NF-pNETs or isolated insulinoma. Laparoscopic pancreatectomies are challenging procedures with potentially severe complications and should be performed in centres with substantial expertise in both laparoscopic and pancreatic surgery. Pancreatic NETs are more often non-aggressive tumours with excellent long-term prognosis and the quality of the tailored surgical strategy remains a major concern for both short- and long-term outcomes.

## Figures and Tables

**Figure 1 jcm-09-02993-f001:**
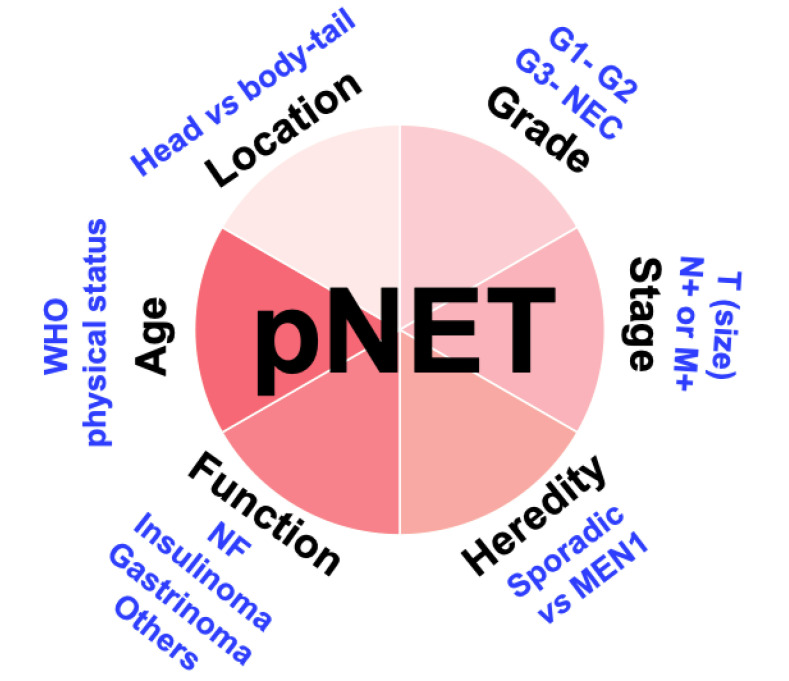
Description of the various factors that should be considered planning the surgical management of patients with pNET.

**Figure 2 jcm-09-02993-f002:**
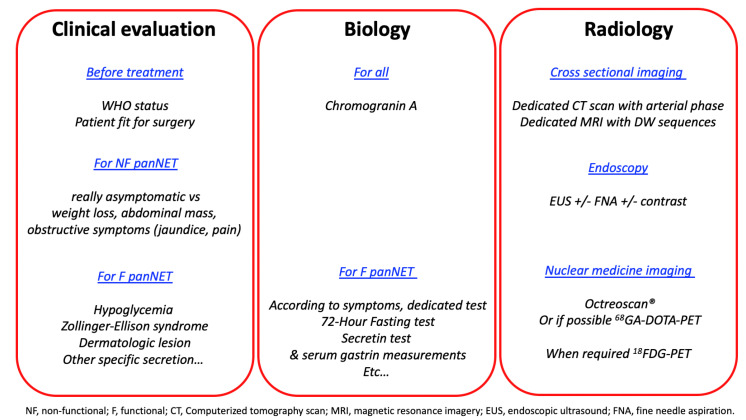
“Diagnose”, “locate“, “stage” and ‘grade” as accurately as possible the tumor.

**Figure 3 jcm-09-02993-f003:**
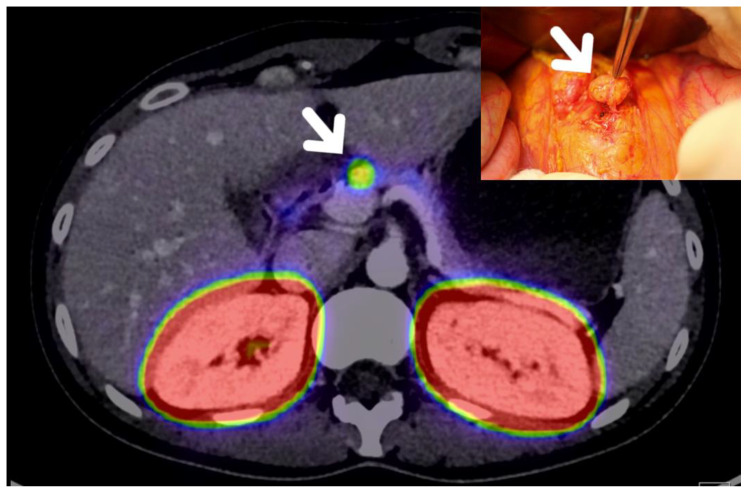
Glucagon-like peptide-1 receptor (GLP-1R) PET/CT locating a 7 mm insulinoma, latter enucleated.

**Figure 4 jcm-09-02993-f004:**
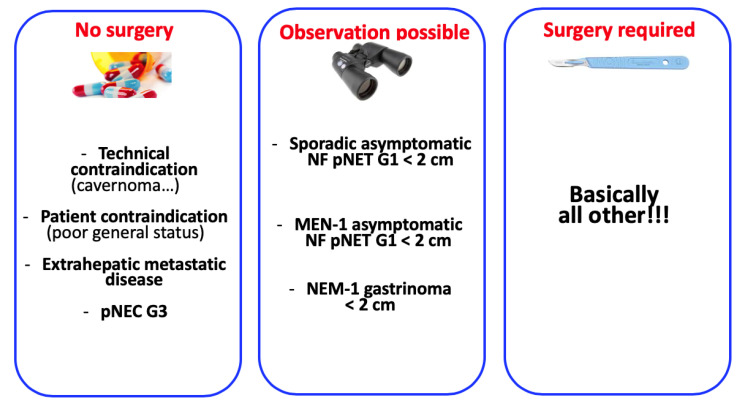
Description of the main surgical indications for pNETs.

**Figure 5 jcm-09-02993-f005:**
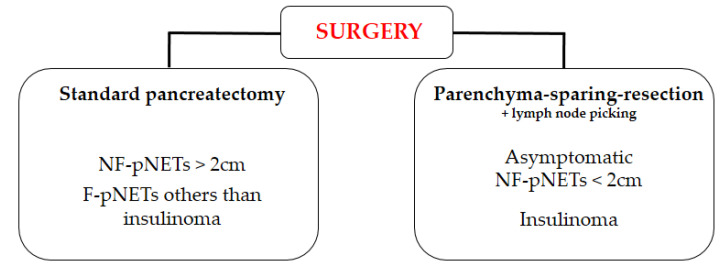
Summary of the indications of standard surgery or pancreatic sparing surgery for pNETs.

**Table 1 jcm-09-02993-t001:** Symptoms, characteristics and surgical approach in relation with the most frequent functional pNET.

Name	Symptoms	Secretion	Incidence New Case//Million/yr.	Location	Malignant	MEN-1 Context	Surgery	Procedure
**Insulinoma**	*Whipple’s triad:* Low blood sugar, presence of symptoms, and reversal of these symptoms when the glucose serum level is restored to normal Many other, like confusion, behavioral changes, visual troubles	insulin	1–32	Variable	<10%	4–5%	Always	Sparing parenchymal pancreatectomy
**Gastrinoma**	*Zollinger-Ellison syndrome*: Gastric acid hypersecretion, severe peptic ulceration, profuse diarrhea	gastrin	0.5–21.5	Stabile & Passaro triangle	60%	20–25%	yes (unless MEN-1 gastrinoma < 2 cm)	Sparing parenchymal or standard pancreatectomy
**Glucagonoma**	Hyperglycemia, necrotic migratory erythema	glucagon	0.01–0.1	Variable	50–80%	1–20%	yes	Sparing parenchymal or standard pancreatectomy
**Vipoma**	*WDHA syndrome* Watery diarrhea, hypokalemia, acidosis	VIP	0.05–0.2	Variable	60–80%	6%	yes	Standard pancreatectomy
**Somatostinoma**	Pain, diabetes, diarrhea, gallstones	somatostatin	<0.02	Variable	70–92%	45%	yes	Standard pancreatectomy

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
