# Peer review of "Surgical Management of Neuroendocrine Tumours of the Pancreas"

_jcm, 2020, doi:10.3390/jcm9092993_

Round 1

Reviewer 1 Report

In the manuscript by Souche et al, the authors provide a review of surgical procedures included in the management of neuroendocrine tumours. Overall, the review is dense with useful information on this topic. However, the publication can be improved by inclusion of more comprehensive figures than ones included in the current version. Here are some suggestions regarding the same:

  1. In section titled “Preoperative evaluation of pNET patients”, three aspects of pe-operative evaluation are described: clinical examination, biological tests and radiological imaging. Include a table summarizing criteria (described in detail in the text) for each evaluation in a bullet point format.
  2. For Figure 2, use an arrow/box to highlight the location of the insulinoma in the bigger figure as it may not be obvious to everyone.
  3. Add a table for different types of different types of pNETs with columns to summarize the detailed explanation in text such as: type, secreted peptide (if any), location, criteria for surgical indication, percentage of total patients etc. This is not an exhaustive list and the authors should include every criterion they feel appropriate.

With these additions, I think the review will be enhanced in its ease of understanding and usefulness to its audience and thus acceptable for publication.

Author Response

Review Report Form 1

In the manuscript by Souche et al, the authors provide a review of surgical procedures included in the management of neuroendocrine tumors. Overall, the review is dense with useful information on this topic.

However, the publication can be improved by inclusion of more comprehensive figures than ones included in the current version. Here are some suggestions regarding the same:

  1. In section titled “Preoperative evaluation of pNET patients”, three aspects of pe-operative evaluation are described: clinical examination, biological tests and radiological imaging. Include a table summarizing criteria (described in detail in the text) for each evaluation in a bullet point format.

We thank the reviewer for his/her comments.

We provide a new table (Table 1) named “Diagnose”, “locate“, “stage” and ‘grade” as accurately as possible the tumor” that summarize the 3 aspects of the preoperative evaluation (clinic, biology and radiology).

Appropriate changes had been made in the manuscript.

  1. For Figure 2, use an arrow/box to highlight the location of the insulinoma in the bigger figure as it may not be obvious to everyone.

We thank the reviewer for his/her comments.

We added two white arrows in the figure 2.

Appropriate changes had been made in the manuscript.

  1. Add a table for different types of different types of pNETs with columns to summarize the detailed explanation in text such as: type, secreted peptide (if any), location, criteria for surgical indication, percentage of total patients etc. This is not an exhaustive list and the authors should include every criterion they feel appropriate.

We thank the reviewer for his/her comments.

We wrote a new table (Table 2) named « Symptoms, characteristics and surgical approach in relation with the most frequent functional pNET” that detail most frequent F-pNETs as you recommended.

Appropriate changes had been made in the manuscript.

With these additions, I think the review will be enhanced in its ease of understanding and usefulness to its audience and thus acceptable for publication.

We thank the reviewer for his/her comments.

Reviewer 2 Report

Manuscript describing management of pancreatic neuroendocrine tumors. Difficult to understand the value of this manuscript, since it does not follow a systematic approach to available data but seems more a standard expert opinion and how we do it publication. The scientific value is therefore low.

The described data is adequate, although the debatable issues ( such as resection of pNEt smaller than 2 cm and/or extensive imaging studies before treatment) could have been discussed from more than the authors perspectives.

If it was a more focussed manuscript, for example the current surgical approaches for pnet, with systematic reviews of available data, it would have had more scientific value.

the issue on dedicated teams and dedicated meetings on patient care strategy are not giving solutions, just a description of current practice.

Author Response

Review Report Form 2

Manuscript describing management of pancreatic neuroendocrine tumors. Difficult to understand the value of this manuscript, since it does not follow a systematic approach to available data but seems more a standard expert opinion and how we do it. The scientific value is therefore low.

The described data is adequate, although the debatable issues (such as resection of pNEt smaller than 2 cm and/or extensive imaging studies before treatment) could have been discussed from more than the authors perspectives.

If it was a more focused manuscript, for example the current surgical approaches for pnet, with systematic reviews of available data, it would have had more scientific value.

the issue on dedicated teams and dedicated meetings on patient care strategy are not giving solutions, just a description of current practice.

This narrative value was asked by the editor for a special issue on pancreatic neuroendocrine tumors in Journal of Clinical Medicine. 

The idea is to propose an overall view of the surgical management of pancreatic neuroendocrine tumor, with a surgical point of view.

According to this specification we wrote this narrative review. If we agree that the scientific value might be low for experts, despite up to date recommendations and references, we believe that its educative value for reader, especially non expert ones, is high.

Reviewer 3 Report

The review "Surgical Management of neuroendocrine tumours of the pancreas"  by Souche et al.

is a well structured and comprehensive paper elucidating the complex context of pNET and surgery.

However, for the sake of completeness some aspects should be discussed addionally:

  1. The option of  endoscopic ultrasound-guided ablation therapy with ethanol (ethanol ablation) should be mentioned
  2. The selective arterial calcium stimulation test (SACST) as a usefull localisation test for insulinom might be important in selected cases. The test is possibly important for the decsision pancreatic head resection vs. distal pancreatectomie. Please discuss.
  3. Regarding minimal-invasiv surgery for neuroendocrine tumours of the pancreas the authors are right: Due to significant increased morbidity / mortality laparoscopic whipple procedure did not prevail. However, robotic pancreatic surgery should be discussed as an alternative.

Author Response

Review Report Form 3

The review "Surgical Management of neuroendocrine tumours of the pancreas" by Souche et al. is a well structured and comprehensive paper elucidating the complex context of pNET and surgery. However, for the sake of completeness some aspects should be discussed additonally:

  1. The option of endoscopic ultrasound-guided ablation therapy with ethanol (ethanol ablation) should be mentioned

We thank the reviewer for his/her comments.

We added the following sentence and subsequent references:

“As an alternative to surgery, endoscopic ultrasound-guided ablation therapies with ethanol or by radiofrequency haverecently been developed, with limited available data to date”

Appropriate changes had been made in the manuscript, and highlighted.

  1. The selective arterial calcium stimulation test (SACST) as a usefull localisation test for insulinoma might be important in selected cases. The test is possibly important for the decsision pancreatic head resection vs. distal pancreatectomie. Please discuss.

We thank the reviewer for his/her comments.

We added the following sentence and subsequent references:

“In the same way, invasive technique is available in case of occult insulinoma as a selective arterial calcium stimulation test (sensitivity 85%) in order to regionalize the insulinoma and plane either a distal pancreatectomy or a pancreaticoduodenectomy. Nevertheless, this technique, at least in Europe, is less and less frequently performed and rarely available even in expert centers. ».

  1. Regarding minimal-invasive surgery for neuroendocrine tumours of the pancreas the authors are right: Due to significant increased morbidity / mortality laparoscopic whipple procedure did not prevail. However, robotic pancreatic surgery should be discussed as an alternative.

We thank the reviewer 3 for this comment. We added the following sentence:

Laparoscopic pancreaticoduodenectomy (LDP) presents some disadvantages compared with conventional laparotomy, including instrument motion with restricted range of movement, two-dimensional imaging, poor surgeon ergonomics and a long learning curve especially to perform pancreatic and biliary anastomosis. Robotic surgery is an advanced minimally invasive surgical technique that has several benefits in complex procedure as pancreaticoduodenectomy, such as enhanced three-dimensional vision, application of EndoWrist instruments that mimics the latter’s hands and consequently, a shorter learning curve compared to standard laparoscopy. Therefore, Robot-assisted pancreaticoduodenectomy (RAPD) might be more advantageous than LPD. The absence of reconstruction required in distal pancreatectomy makes robot-assisted distal pancreatectomy less attractive and less cost effective, especially for benign or borderline lesion as pNETs (90, 91). Recent reports have shown that RAPD is safer and more efficient than LPD among properly selected patients (92-94). Therefore, RAPD is technically a feasible alternative to the laparoscopic procedure. Further studies may be needed to evaluate the cost-effectiveness of RAPD (94).

Otherwise, the results appear more encouraging for minimally invasive pancreatectomies that do not require reconstructions, explaining that laparoscopic enucleation and distal pancreatectomy were widely adopted.

Reviewer 4 Report

This is a narrative review regarding a multidisciplinary approach to pNET with a focus about surgical treatment. 

I have only minor suggestions:

  • there are some typos to be corrected (for example "stagging" p4line109, NEM-1 in figure 3)
  • "While surgical resection is systematically indicated in patients with functional pNET and NF-pNET ≥2 cm, a surveillance strategy is recommended in patients with small NF-pNET or small non metastatic MEN1 gastrinoma." please provide a reference

  • "As an alternative to surgery, radiofrequency ablation has recently been developed, with limited available data to date." please provide a reference
  • I suggest to add a small chapter about follow-up, risk of recurrence and survival 

  • The list of references should be enriched 

Author Response

Review Report Form 4

This is a narrative review regarding a multidisciplinary approach to pNET with a focus about surgical treatment. I have only minor suggestions:

There are some typos to be corrected (for example "stagging" p4line109, NEM-1 in figure 3)

We thank the reviewer 3 for this comment. These errors were corrected.

"While surgical resection is systematically indicated in patients with functional pNET and NF-pNET ≥2 cm, a surveillance strategy is recommended in patients with small NF-pNET or small non metastatic MEN1 gastrinoma." please provide a reference

We thank the reviewer 3 for this comment.

We provide adequate references about this point.

"As an alternative to surgery, radiofrequency ablation has recently been developed, with limited available data to date." please provide a reference

We thank the reviewer 3 for this comment.

We provide adequate references about this point.

I suggest to add a small chapter about follow-up, risk of recurrence and survival 

We thank the reviewer 3 for this comment.

We added the following paragraphs:

Survival:

“According to the American Cancer Society (cancer.org) relying on information from the SEER* database between 2009 and 2015, the 5-year survival rate for resected patient with non-metastatic pNETs is 93%. In locally advanced pNETs or positive regional lymph nodes, the 5-year survival rate is 77%. In metastatic patients, the 5-years survival rate is 27%.”

Recurrence and Follow-up

Despite a 5-year survival rate for resected patient is 93%, recurrence rate is 12%–25% at 5 years and late recurrence were described (up to 70% at 15 years). Follow-up requires a close monitoring with physical examination, biology (essentially for F-pNETS) and cross-sectional imaging (CT/MRI), 3–6 monthly for NF-pNETs and 6–12 monthly for F-pNETs every year for 3 years, every 1–2 years thereafter until 10 years post-pancreatectomy. Although insulinoma may not require any radiological follow-up, ENETS guidelines recommend a life-long surveillance for all other pNETs.”

The list of references should be enriched 

We thank the reviewer 3 for this comment. Referenced were enr

Round 2

Reviewer 2 Report

Revisions were made and they improved the manuscript considerably; especially the text on options for specific lesions.